# Low Influenza, Pneumococcal and Diphtheria–Tetanus–Poliomyelitis Vaccine Coverage in Patients with Primary Sjögren’s Syndrome: A Cross-Sectional Study

**DOI:** 10.3390/vaccines8010003

**Published:** 2019-12-21

**Authors:** Morel Jacques, Hind Letaief, Guilpain Philippe, Mariette Xavier, Combe Bernard, Cédric Lukas

**Affiliations:** 1Department of Rheumatology, CHU and Montpellier University, 34295 Montpellier, France; h-letaief@chu-montpellier.fr (H.L.); b-combe@chu-montpellier.fr (C.B.); c-lukas@chu-montpellier.fr (C.L.); 2Department of Internal Medicine and Multi-Organic Diseases, CHU and Montpellier University, 34295 Montpellier, France; p-guilpain@chu-montpelllier.fr; 3AP-HP, Université Paris-Saclay, Department of Rheumatology, University of Paris Sud, INSERM, UMR1184, Center of Immunology of Viral Infections, Auto-Immune Diseases, 94270 Le Kremlin Bicêtre, France; xavier.mariette@aphp.fr

**Keywords:** Sjögren’s syndrome, vaccination, coverage

## Abstract

*Objective:* To evaluate vaccination coverage and reasons for non-vaccination in patients with primary Sjögren’s syndrome (pSS). *Method:* A total of 111 patients fulfilling American–European Consensus Group criteria for pSS were interviewed by use of a standardized questionnaire between January 2016 and November 2017 in two French tertiary referral centers for auto-immune diseases. *Results:* Updated immunization coverage for influenza was 31.5% (*n* = 35), pneumococcus was 11.7% (*n* = 13), and diphtheria–tetanus–poliomyelitis (DTP) was 24.3% (*n* = 27). The main reasons for non-vaccination were fear of side effects from the influenza vaccine (40.3%) and a lack of proposal for the pneumococcal vaccine (72.3%). In vaccinated patients, vaccination was mainly proposed by general practitioners for the influenza vaccine (42.6%) and rheumatologists for the pneumococcal vaccine (41.2%). Probability of influenza vaccination was associated with age (odds ratio/year (OR) 1.04, 95% confidence interval (CI) 1.0–1.1; *p* = 0.016), history of severe infection (OR 15.9, 95% CI 1.35–186; *p* = 0.028), low EULAR Sjögren’s syndrome disease activity index (OR 0.85, 95% CI 0.75–0.96; *p* = 0.013), and comorbidities (OR 3.52, 95% CI 1.22–10.2; *p* = 0.02). Probability of vaccination against pneumococcus was associated with lung comorbidities (OR 3.83, 95% CI 1.11–13.12; *p* = 0.033) and up-to-date influenza vaccination (OR 3.71, 95% CI 1.08–12.8; *p* = 0.038). *Conclusion:* Influenza, pneumococcal, and DTP vaccine coverage was low in patients with pSS included in this study. These results underline the relevance of systematically screening vaccine status in pSS patients and educating patients and physicians on the need for vaccination to improve vaccine coverage in this population.

## 1. Introduction

Sjögren’s syndrome (pSS) is an autoimmune disease present in approximately one third of patients characterized by lymphoid infiltration of the exocrine glands, especially the salivary and lacrimal glands which are responsible for mouth and eye dryness, and by potential systemic disease manifestations [1]. Even today, pSS management is based mainly on the symptomatic treatment of different mucosal disorders. In patients with systemic manifestations, such as joint, muscular, neurological, pulmonary, or renal involvement, immunomodulatory or immunosuppressive treatments may be considered, even though a drug with clear demonstrated efficacy for these manifestations does not exist.

Infections have been described as one of the major causes of death in pSS [2,3,4]. This susceptibility to infections in autoimmune diseases, including pSS, is due to intrinsic causes related to the disease and also to immunosuppressive treatments. This inherent risk of infections in autoimmune disease could be related to an intrinsic immune deficiency, which could be explained by the immunosenescence or premature aging of the immune system [5] and leukopenia, both of which are common in pSS [2,6]. Indeed, in a systematic meta-analysis review, Brito-Zeron et al. reported that among the variables present at the time of the initial diagnosis of pSS, lymphopenia (1000/mm^3^) was the main risk factor for mortality related to infection, with a relative risk of 4.08 (95% confidence interval (CI) 1.67–9.94) [2]. In a prospective cohort study, the risk of hospitalization for infection was higher in patients with neutropenia than in those with normal polynuclear neutrophil counts (24% vs. 9%, *p* = 0.002) [6].

Other intrinsic factors of infectious risk, particularly pulmonary, have been reported in pSS. Indeed, abnormalities in mucociliary clearance and bronchiectasis, which are frequently present in pSS, are also involved in this increased risk of infection [7,8]. The prevalence of bronchiectasis in patients with pSS ranges from 22%–54%, as observed from high-resolution CT imaging [9,10,11]. These patients are more vulnerable to respiratory tract infections [8].

Immunosuppressive treatments, such as synthetic or biological disease-modifying anti-rheumatic drugs (bDMARDS) [12] or oral corticosteroids, increase the risk of infections in patients with autoimmune diseases, while hydroxychloroquine has a reported protective effect [13,14,15]. Therefore, exposure to these treatments may increase the risk of severe infections in patients with pSS.

To prevent infection, two vaccinations are recommended for immunocompromised patients, i.e., the influenza vaccine and the pneumococcal vaccine [16,17]. Recommendations for the diphtheria–tetanus–poliomyelitis (DTP) vaccine vary across the institutions from which they originate. Indeed, the European recommendations issued by EULAR for this vaccination are identical to those applicable to the general population [17]. According to French recommendations, the DTP booster should be performed every 10 years in all patients with autoimmune diseases [16].

Despite these recommendations, many studies reported that vaccination coverage in patients with chronic inflammatory diseases, such as rheumatoid arthritis (RA), spondyloarthritis, or systemic sclerosis, is very low. To the best of our knowledge, there are no data regarding vaccine coverage in patients with pSS. In this study, we evaluated vaccination coverage for influenza, pneumococcus, and DTP in patients with pSS and investigated the reasons for non-vaccination.

## 2. Patients and Methods

A cross-sectional study was performed in pSS patients from two different French tertiary referral centers for autoimmune diseases (Paris–Bicêtre and Montpellier). From January 2016 to November 2017, questionnaires were randomly delivered to patients with pSS according to European–American Diagnostic Criteria (2002). Before completing the questionnaire, patients gave their consent to participate. This questionnaire was adapted from questionnaires used by the French national agency “Institut de Veille Sanitaire” to study vaccination coverage and were completed with the assistance of one fellow (HL) to limit missing data [18]. The appropriate Institutional Review Board (Comité de Protection des personnes Sud-Mediterranée III) approved the study protocol (register: 2019_IRB-MTP_12–28) and, based on the observational design, waived the need for written informed consent.

Data collected in the questionnaire included previous vaccinations, reasons for non-vaccination, sources of vaccine proposition, and sociodemographic data, including education level (Bachelor degree and post-Bachelor degree education) and the presence of young child(ren) (<10 years old) at home. The following data were collected from the medical file: European–American Diagnostic Criteria (2002) for pSS, the most recent EULAR Sjögren’s syndrome disease activity index (ESSDAI), comorbidities (chronic lung disease, diabetes, chronic kidney disease, chronic liver disease, chronic heart disease, cardiovascular comorbidities (coronary or cerebral ischemia) and severe neurological or muscle disease), history of severe infection (requiring intravenous antibiotics or hospital admission), current smoking status, and treatments used for pSS, including hydroxychloroquine, immunosuppressive drugs (methotrexate, leflunomide, ciclosporine, azathioprine, mycophenolate mofetil), and biological disease-modifying anti-rheumatic drugs (bDMARDs).

For descriptive statistical analysis, the mean ± SD were used for continuous variables and frequencies (%) were used for categorical variables. To evaluate the factors associated with up-to-date vaccination, we compared categorical variables between patients with up-to-date and non-up-to-date influenza, pneumococcal, or DTP vaccination using Fisher’s exact test or the chi-square test as appropriate. Continuous variables (age, ESSDAI) were compared using Student’s *t* test. A binary logistic regression model was used for the multivariate analysis to estimate the odds ratio (OR) of vaccination with a 95% confidence interval (CI). Statistical significance was considered to be *p* < 0.05. All statistical analyses were performed using IBM SPSS 15 software.

## 3. Results

The main clinical and sociodemographic characteristics of patients are described in Table 1. We included 111 patients with pSS, who were mainly women (94%; *n* = 104). The mean age at inclusion was 57 ± 15 years. Half of the patients (53.2%) were positive for anti-SSA antibody and the mean ESSDAI score was 5.1 ± 5.5. About half of the patients (56.4%) had a history of infection within the past five years, including influenza, pneumonia, or severe infection. Only 15.3% (*n* = 17) reported vaccination once in their life against pneumococcus, 48.6% (*n* = 54) received the influenza vaccine during their lifetime, and 9.9% (*n* = 11) received both vaccines in their lifetime.

Immunization coverage for influenza was 31.5% (*n* = 35) within the year preceding the date of questionnaire completion (defining updated influenza vaccination). Updated pneumococcal (≤5 years) and DTP (≤10 years) vaccine coverage were 11.7% (*n* = 13) and 24.3% (*n* = 27), respectively.

In vaccinated patients (*n* = 54), the influenza vaccine was proposed by a general practitioner for 42.6% (*n* = 23) of patients, a rheumatologist for 11.1% (*n* = 6), health insurance for 24.1% (*n* = 13), and another source for 29.6% (*n* = 16).

A total of 57 patients reported that they had never been vaccinated against influenza during their lifetime. Among these patients, the reported reasons were the fear of side effects for 40.3% of patients (n = 23) and a lack of proposal for vaccination by a healthcare professional for 28.1% (*n* = 16). Overall, 21.4% (*n* = 12) of patients reported the use of a method supported by alternative medicine users for vaccination (e.g., homeopathic vaccine, essential oils, herbal medicine). Six patients mentioned another (undefined) reason for non-vaccination. In total, 17.9% (*n* = 10) did not feel they were at risk of infection and 8.8% (*n* = 5) considered the vaccine to be contraindicated in their disease, according to their treating physician. Four patients considered the vaccine to be contraindicated because of their treatment, according to their treating physician.

Only 13 patients (11.7%) were correctly vaccinated against pneumococcus. The pneumococcal vaccination was proposed by a rheumatologist for 41.2% (*n* = 7) of patients, a general practitioner for 29.4% (*n* = 5), a lung specialist for 17.6% (*n* = 3), an occupational physician for 5.9% (*n* = 1), or another source for 5.9% (*n* = 1). A total of 94 patients reported that they had never been vaccinated against pneumococcus during their lifetime. Among the patients not vaccinated against pneumococcus, the reasons for non-vaccination were a lack of proposal of vaccination by a healthcare professional, cited by 72.3% (*n* = 68) of patients, and the fear of side effects by 16% (*n* = 15). Eleven patients reported another reason for non-vaccination. Overall, 6.3% (*n* = 6) of patients considered themselves to not be at risk of infection and 2.1% (*n* = 2) thought that the vaccine was contraindicated because of their pSS status, according to the treating physician.

We compared the factors associated with up-to-date vaccination status for the influenza vaccine (Table 2), the pneumococcal vaccine (Table 3), and the DTP vaccine (Table 4). The 35 patients who were up-to-date with their influenza vaccines were significantly older, had a more frequent history of severe infection, and received other vaccine boosters more frequently. The 13 patients who were up-to-date with their pneumococcal vaccines kept up-to-date with their influenza vaccines more frequently and also experienced lung disease more often. Patients with up-to-date DTP vaccination statuses were younger than those without up-to-date DTP vaccination (mean 48.4 ± 16.4 vs. 60.4 ± 12.9 years, *p* < 0.001).

As observed from the logistic regression analysis, factors associated with updated influenza vaccine were age (OR 1.04/year, 95% CI 1.0–1.1; *p* = 0.016), a history of severe infection (OR 15.9, 95% CI 1.35–186 *p* = 0.028), a low ESSDAI activity score (OR 0.85, 95% CI 0.75–0.96; *p* = 0.013), and the presence of at least one comorbidity (OR 3.52, 95% CI 1.22–10.2; *p* = 0.02). Factors associated with vaccination against pneumococcus were lung comorbidities (OR 3.83, 95% CI 1.11–13.2; *p* = 0.033) and up-to-date influenza vaccination (OR 3.71, 95% CI 1.08–12.8; *p* = 0.038). Factors associated with DTP vaccination were age (OR 0.92/year, 95% CI 0.89–0.96; *p* < 0.0001), a history of severe infection (OR 7.79, 95% CI 1.10–55.3 *p* = 0.04), and the use of glucocorticoids (either past or current) (OR 3.95, 95% CI 1.15–13.6; *p* = 0.03).

## 4. Discussion

In our patients with pSS, we observed low vaccination coverage for influenza (31.5%), pneumococcus (11.7%), and DTP (24.3%), despite current consensual recommendations advising systematic vaccination. In the French general population, coverage for the DTP and pneumococcal vaccines administered within 10 years was 44% and 8%, respectively, in healthy adults aged ≥65 years old [18]. In 2017–2018, the coverage in healthy adults for the influenza vaccine was 29% for those aged <65 years old and 50% for those aged ≥65 years old. In pSS patients, vaccination coverage rates were closer to the rates observed in the general population than those reported in other rheumatic diseases, especially for the pneumococcal vaccine. Indeed, in rheumatic diseases such as RA, influenza vaccine coverage was 28%–55% and pneumococcal vaccine coverage was 28%–62% [19,20]. In a French study performed in 2005, influenza vaccine coverage was 28% among 137 patients with autoimmune and chronic inflammatory diseases who received immunosuppressive therapy [19]. More recently, we conducted a study in patients with chronic inflammatory rheumatic diseases with a similar methodology. In 454 patients with RA or spondyloarthritis, pneumococcal vaccine coverage was 53.6% and influenza vaccine coverage was 54.5% [20]. In this study, 89% of patients with RA had already received at least one bDMARD and 87.8% of those with spondyloarthritis had already received anti-tumor necrosis factor α. Usually, patients receiving bDMARDs benefit from a pre-therapeutic assessment, including a vaccination update. bDMARDs are not used as often in pSS as in other chronic inflammatory or autoimmune diseases, which could explain these low rates of vaccine coverage. Indeed, in our study, only 20.7% (*n* = 23) of patients reported treatment with a bDMARD in their lifetime. This lower use of bDMARDs compared to other studies conducted on RA and SpA may explain the lower vaccination coverage for influenza and pneumococcus but does not explain the low immunization coverage for DTP. This rate was similar to that reported in a recent study in Germany of 331 patients with RA, where 43% of patients received an anti-tetanus booster in the previous 10 years, and 26% and 16.9% were up-to-date for their diphtheria and polio vaccines, respectively [21]. In this study, the rate of vaccine coverage was significantly better for older than younger patients. The mean age of patients vaccinated against influenza was 61.1 ± 14.2 versus 54.5 ± 14.2 years for non-vaccinated participants (*p* = 0.01). Conversely, younger patients exhibited better immunization coverage for DTP than older patients (33.9% vs. 6.8%, *p* < 0.0001). This better coverage rate of the influenza vaccine in older than younger patients may be explained by the specific indication for vaccination in this age group in the general population. Indeed, in France, people over 65 years old automatically receive a letter from the health insurance inviting them to be vaccinated against influenza at no cost to themselves.

The overall lower vaccination coverage rates for patients with pSS than other autoimmune diseases could also be explained by the fact that healthcare professionals seem less aware of the importance of vaccination, especially the pneumococcal vaccine, in pSS. Indeed, in our study, 79.1% of non-vaccinated patients declared that this vaccine was not proposed to them by healthcare professionals as the main reason for non-vaccination. Lack of proposal was the second most common reason mentioned for the influenza vaccine (28.6% among non-vaccinated patients), after fear of side effects. Data showing the clinical effectiveness of vaccination are scarce, especially in immunocompromised patients; this low level of evidence may also explain the low prescription of pneumococcal vaccine in this population. Only one randomized controlled trial investigating the efficacy of pneumococcal vaccine in patients with pSS was performed, which included 32 patients with pSS who were not receiving immunosuppressive drugs being vaccinated with the 14-valent pneumococcal PS vaccine (PPV14). A response to PPV14 was observed after one month, but decreased at six months after vaccination [22].

Lastly, a fear of side effects was the main reason observed for non-vaccination against influenza in our study (40.3% among non-vaccinated patients). This was in-line with the rise in vaccine hesitancy observed in France over the past 10 years [23]. The general low adherence of the French population to vaccination may be explained by a growing distrust in the safety of vaccines and their components, like aluminum adjuvants [23]. This fear, which is not justified in the general population, could be more understandable in autoimmune diseases. However, all the available data and the recently reviewed EULAR recommendations are very reassuring and do not suggest a significant risk of relapse in patients with autoimmune diseases [17]. In Sjögren’s syndrome, increases in IgG, anti-SSA, and anti-SSB titers patients’ sera after influenza vaccination without any increase in flare-ups were recently reported [24].

A low disease activity score (ESSDAI in our study) was associated with up-to-date influenza vaccination, which was also previously observed in 3920 RA patients [25]. Indeed, in the COMORA study conducted in 17 countries, optimal influenza vaccination was associated with low disease activity defined by a Disease Activity Score of <3.2 in 28 joints (OR 1.37, 95% CI 1.13–1.65). This finding was in agreement with the EULAR recommendations, suggesting that patients with autoimmune and rheumatic diseases should ideally be vaccinated when the disease is stable [17].

The presence of comorbidities, such as a history of severe infection, lung comorbidities, or presence of at least one comorbidity, was associated with up-to-date influenza or pneumococcal vaccination. The presence of one or more comorbidities, especially chronic lung disease, was associated with influenza and pneumococcal vaccination. Haroon et al. also found that chronic lung disease was associated with influenza and pneumococcal vaccination, with OR values of 8.62 (95% CI 2.2–33.8; *p* = 0.002) and 4.94 (95% CI 1.24–19.7; *p* = 0.023) [26]. The presence of lung comorbidities or a previous severe infection likely warned the physician of the risk of lung infection, thereby resulting in a higher likelihood of proposing these vaccines to prevent influenza or pneumococcal infections. We observed a reciprocal association between up-to-date influenza and pneumococcal vaccinations, which supported this hypothesis.

Our study has some limitations. Data were self-reported by patients and therefore sensitive to recall bias. The analysis of pneumococcal vaccination coverage was non-systematic because it did not distinguish between patients who had the two recommended vaccine doses recommended for complete coverage and those who had incomplete vaccination. The collection and analysis of data regarding corticosteroids and immunosuppressive treatments were partial because we were unable to identify doses or durations of corticosteroid therapy or distinguish between current and former treatments. This was a French bicentric study that may not reflect vaccine coverage in all countries. However, the two referral centers for autoimmune diseases propose the collection of comorbidities, including vaccine status, for patients with pSS, and rates of vaccine coverage are probably in the higher range. Because of the small sample size, it is possible that many of the negative findings were due to insufficient numbers to detect differences between up-to-date and non-up-to-date patients. In addition, our population was mainly composed of women (93%), as observed in an independent French cohort of pSS (ASSESS) [27]. In other cohorts, men were more represented, which could partly explain the low vaccination coverage in our population [28], as men usually experience more severe disease with shorter follow-up intervals. It is known that patients who are monitored closely have better vaccination coverage than patients who are managed with no specific instructions [29].

## 5. Conclusions

This study showed that the coverage levels of the influenza, pneumococcal, and DTP vaccines are low in patients with pSS. It also emphasized the relevance of systematically screening vaccine status in patients with pSS and the importance of educating patients and physicians regarding the need for these vaccinations to improve vaccine coverage in this population.

## Figures and Tables

**Table 1 vaccines-08-00003-t001:** Characteristics of patients (n = 111).

Characteristics	(n = 111)
Age (Mean ± SD)	57 ± 15
Gender: women (%)	104 (93.7)
Minor Salivary Gland Biopsy FS ≥ 1 (%)	79 (70.8)
Anti-SSA positive (%)	59 (53.2)
ESSDAI (mean ± SD)	5.1 ± 5.5
Current smokers (%)	20 (18)
Young child(ren) (<10 years old) at home (%)	14 (12.6)
Bachelor degree (%)	52 (46.8)
Comorbidity ≥ 1 (%)	68 (61.3)
Steroids (%)	74 (66.6)
Hydroxychloroquine (%)	73 (65.8)
IS used ≥ 1	82 (73.9)
Methotrexate (%)	42 (37.8)
biological DMARD used ≥ 1 (%)	24 (21.6)

SD = standard deviation; ESSDAI; EULAR Sjögren’s Syndrome Disease Activity Index; FS = Focus Score; IS = immuno-suppressive drugs (methotrexate, leflunomide, ciclosporine, azathioprine, mycophenolate mofetil); bDMARD = biological disease-modifying anti-rheumatic drug.

**Table 2 vaccines-08-00003-t002:** Factors associated with updated (≤1 year) influenza vaccine status.

	Updated (n = 35)	Not Updated (n = 75)	*p*
Age (mean ± SD)	62.7 ± 14.7	55.4 ± 13.9	0.01
Age ≥ 65 years(%)	16/35 (45.7)	22/75 (29.3)	0.09
bDMARD used ≥ 1 (%)	6/35 (17.1)	18/75 (24)	0.4
Comorbidity ≥ 1 (%)	19/35 (54.3)	9/75 (38.7)	0.1
Current smokers (%)	5/32 (15.6)	15/72 (20.8)	0.5
ESSDAI (mean ± SD)	3.7 ± 4.5	5.7 ± 5.8	0.09
Gender: women (%)	34/35 (97.1)	69/75 (92)	0.4
Bachelor degree (%)	14/33 (42.4)	38/71 (53.5)	0.29
History of severe infection (%)	7/35 (20)	1/75 (1.3)	0.001
Immuno-suppressive drug used ≥ 1 (%)	28/35 (80)	53/75 (70.7)	0.4
Lung comorbidity (%)	11/35 (31.4)	16/75 (21.3)	0.2
Steroids (%)	23/35 (65.7)	46/75 (61.3)	0.6
Vaccination update > 1 vaccine (%)	28/35 (80)	40/75 (53.3)	0.007

Binary logistic regression model. SD = standard deviation; ESSDAI; Eular Sjögren’s Syndrome Disease Activity Index; bDMARD = biological disease-modifying anti-rheumatic drug.

**Table 3 vaccines-08-00003-t003:** Factors associated with updated (≤5 years) pneumococcal vaccine status.

	Updated (n = 13)	Not Updated (n = 98)	*p*
Age (mean ± SD)	59.1 ± 11.7	57.3 ± 15.1	0.7
Age ≥ 65 years (%)	3/13 (23.1)	35/98 (35.7)	0.5
Gender: women (%)	13/13 (100)	91/98 (92.9)	1
ESSDAI (mean ± SD)	6.33 ± 5.7	5.47 ± 4.9	0.4
Young child(ren) at home (%)	(3/13 (23.1)	11/93 (11.8)	0.4
Current smokers (%)	2/12 (16.7)	18/93(19.4)	1
up-to-date flu vaccine (%)	8/13 (61.5)	27/97 (27.8)	0.02
History of severe infection (%)	3/13 (23.1)	5/98 (5.1)	0.05
Steroids (%)	10/13 (76.9)	59/98 (60.2)	0.4
IS drug used ≥ 1 (%)	11/13 (84.6)	70/98 (71.4)	0.5
bDMARD used ≥ 1 (%)	4/13 (30.8)	20/98 (20.4)	0.5
Vaccination update ≥ 1 vaccine (%)	10/13 (76.9)	58/98 (59.2)	0.2
Comorbidity ≥ 1 (%)	8/13 (61.5)	40/98 (40.8)	0.2
Lung comorbidity (%)	7/13 (53.8)	20/98 (20.4)	0.01
Bachelor degree (%)	8/13 (61.5)	44/92 (47.8)	0.4

Binary logistic regression model. SD = standard deviation; ESSDAI; Eular Sjögren’s Syndrome Disease Activity Index; Immuno-suppressive drugs (methotrexate, leflunomide, ciclosporine, azathioprine, mycophenolate mofetil); bDMARD = biological disease-modifying anti-rheumatic drug.

**Table 4 vaccines-08-00003-t004:** Factors associated with updated (≤10 years) diphtheria-tetanus-poliomyelitis vaccine status.

	Updated (n = 27)	Not Updated (n = 84)	*p*
Age (mean ± SD)	48.4 ± 16.4	60.4 ± 12.9	<0.001
Age ≥ 65 years (%)	4/27 (14.8)	34/84 (40.5)	0.01
Gender: women (%)	25/27 (92.6)	79/84 (94)	0.7
ESSDAI (mean ± SD)	5.41 ± 5.86	4.97 ± 5.38	0.7
Young children (<10 years old) at home (%)	5/26 (19.2)	9/80 (11.3)	0.3
Current smokers (%)	6/25 (24)	14/80 (17.5)	0.6
Up-to-date Flu vaccine (%)	9/27 (33.3)	31.3 (26/83)	0.8
History of severe infection (%)	4/27 (14.8)	4/84 (4.8)	0.1
Steroids (%)	20/27 (74.1)	49/84 (58.3)	0.1
IS drug used ≥ 1 (%)	22/27 (81.5)	59/84 (70.2)	0.25
bDMARD drug used ≥ 1 (%)	8/27 (29.6)	16/84 (19)	0.24
Comorbidity ≥ 1 (%)	11/27 (40.7)	37/84 (44)	0.8
Lung comorbidity (%)	8/27 (29.6)	19/84 ((22.6)	0.4
Bachelor degree (%)	17/26 (65.4)	(35/79 (44.3)	0.06

Binary logistic regression model. SD = standard deviation; ESSDAI; Eular Sjögren Syndrome Disease Activity Index; IS = immuno-suppressive drugs (methotrexate, leflunomide, ciclosporine, azathioprine, mycophenolate mofetil); bDMARD = biological disease-modifying anti-rheumatic drug.

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
