# Peer review of "Low Influenza, Pneumococcal and Diphtheria–Tetanus–Poliomyelitis Vaccine Coverage in Patients with Primary Sjögren’s Syndrome: A Cross-Sectional Study"

_vaccines, 2019, doi:10.3390/vaccines8010003_

Round 1
Reviewer 1 Report
A major limitation of this study is the small sample size. It is possible that many of the negative findings are due to insufficient power to detect differences between updated and not updated as presented in tables 2 and 3.
The investigators need to compare the vaccination coverage for pSS patients to the French general population. The low vaccine coverage in pSS patients may be a reflection of the vaccination coverage of the general population.
The manuscript needs text editing to clarify some terms like "Graduated scholar level" and "The data were declarative".
Author Response
We thank the referees for their pertinent comments and we answered point by point the questions raised by them. Please please find the responses to reviewers and the corrected manuscript entitled Low influenza, pneumococcal and diphtheria-tetanus-poliomyelitis vaccine coverage in patients with primary Sjögren’s syndrome: a cross-sectional study” by Morel, al, we would like to submit for consideration in Vaccines journal. We hope that our manuscript is now ready for publication in your journal. In the corrected manuscript attached, corrections are highlighted in red letters.
Referee (1) Comments to the Author:
A major limitation of this study is the small sample size. It is possible that many of the negative findings are due to insufficient power to detect differences between updated and not updated as presented in tables 2 and 3.
Thank you for this comment. Indeed, the sample size is a limitation. We added in the limitations of our study this sentence (line 239-242): Because of the small sample size, it is possible that many of the negative findings are due to insufficient power to detect differences between updated and not updated patients.
The investigators need to compare the vaccination coverage for pSS patients to the French general population. The low vaccine coverage in pSS patients may be a reflection of the vaccination coverage of the general population.
Thank you for this comment. We compared the vaccination coverage for SS patients with the French general population. In the French general population, coverage for DTP vaccine administrated within 10 years was 44%, and 8% for pneumococcal vaccine, in healthy adults aged ≥65 years old. In 2017-2018, for influenza vaccine, the coverage in healthy adults was 29% for age <65 years old and 50% for age≥65 years old. In pSS patients, vaccination coverage rates are closer to rates observed in general population than those reported in other rheumatic diseases, especially for the pneumococcal vaccine.
(Reference:Guthmann JP,Fonteneau L, Bonmarin I ,Lévy Bruhl D,. Institut de veille sanitaire – Enquête nationale de couverture vaccinale, France, janvier 2011. We added this paragraph and references line 164-169.
The manuscript needs text editing to clarify some terms like "Graduated scholar level" and "The data were declarative".
We changed Graduated scholar for Bachelor degree since the question was : Do you have Bachelor Degree? We changed this information line 91-92.
We replaced the word declarative by “Data were self reported by patients and therefore sensitive to recall bias.”

Reviewer 2 Report
The manuscript titled “Low influenza, pneumococcal and diphtheria-tetanus-poliomyelitis vaccine coverage in patients with primary Sjögren’s syndrome: a cross-sectional study” investigates the rate of vaccination among the primary Sjögren’s syndrome patients and the reasons for not having the vaccinations. Influenza and Pneumococcus vaccinations are recommended for the affected patients. However, the vaccination rate was only 31.5 and 11.7% respectively. The analysis reveals some interesting factors such as how age influences influenza and pneumococcus vaccination rates and the percentage of patient population that were not advised for vaccinations by health professionals. This information is important to adopt better strategies to increase the vaccination rate among primary Sjögren’s syndrome patients and other autoimmune diseases. However, there are some concerns, that should be addressed which can improve the value of the manuscript.
Major concerns:
Is there any information available on the vaccination rate of influenza, pneumococcus, and DTP vaccines among the general population. I would suggest to include this information for comparative purposes. Lines 147-152 enumerates the factors that are associated with influenza and pneumocococcus vaccinations. Did the authors find any similar factors against DTP vaccination? The sample population included more than 94% women. Ramírez Sepúlveda JI et. al. found increased frequency of extra glandular involvement in men, with pulmonary involvement, vasculitis, and lymphadenopathy and also found an increased anti-Ro52 level compared to females. They also concluded that pSS in men manifested more disease severity, even though there is a lower risk of pSS development in men (doi:10.1186/s13293-017-0137-7). Having said that, the authors have not discussed how this gender bias in samples, might have affected this study and also the reasons for the low sample number for men. This could be discussed among the limitations of the study that authors have already listed.Minor concerns:
All tables are of low resolution. Also, make the formatting for all tables uniform.Author Response
We thank the referees for their pertinent comments and we answered point by point the questions raised by them. Please please find the responses to reviewers and the corrected manuscript entitled Low influenza, pneumococcal and diphtheria-tetanus-poliomyelitis vaccine coverage in patients with primary Sjögren’s syndrome: a cross-sectional study” by Morel, al, we would like to submit for consideration in Vaccines journal. We hope that our manuscript is now ready for publication in your journal. In the corrected manuscript attached, corrections are highlighted in red letters.
Referee (2) Comments to the Author:
Is there any information available on the vaccination rate of influenza, pneumococcus, and DTP vaccines among the general population. I would suggest to include this information for comparative purposes.
Thank you for this comment. We compared the vaccination coverage for SS patients with the French general population. In the French general population, coverage for DTP vaccine administrated within 10 years was 44%, and 8% for pneumococcal vaccine, in healthy adults aged ≥65 years old. In 2017-2018, for influenza vaccine, the coverage in healthy adults was 29% for age <65 years old and 50% for age≥65 years old. In pSS patients, vaccination coverage rates are closer to rates observed in general population than those reported in other rheumatic diseases, especially for the pneumococcal vaccine. (Reference:Guthmann JP,Fonteneau L, Bonmarin I ,Lévy Bruhl D,. Institut de veille sanitaire – Enquête nationale de couverture vaccinale, France, janvier 2011. We added this paragraph and references line 164-169
Lines 147-152 enumerates the factors that are associated with influenza and pneumocococcus vaccinations. Did the authors find any similar factors against DTP vaccination?
Thank you for this comment. Indeed, we had observed similar factors for DTP vaccination, which we had first chosen not to report since there are no specific recommendations regarding these vaccines as compared with general population. Howevere, since the reviewer raises the point, we have added these data as follows line 155-157: “Factors associated with DTP vaccination were age (OR 0.92/year, 95% CI 0.89-0.96; p< 0.0001), a history of severe infection (OR 7.79, 95% CI 1.10-55.3 p= 0.04) and the use of glucocorticoids (either past or current) (OR 3.95, 95% CI 1.15-13.6; p= 0.03).”
The sample population included more than 94% women. Ramírez Sepúlveda JI et. al. found increased frequency of extra glandular involvement in men, with pulmonary involvement, vasculitis, and lymphadenopathy and also found an increased anti-Ro52 level compared to females. They also concluded that pSS in men manifested more disease severity, even though there is a lower risk of pSS development in men (doi:10.1186/s13293-017-0137-7). Having said that, the authors have not discussed how this gender bias in samples, might have affected this study and also the reasons for the low sample number for men. This could be discussed among the limitations of the study that authors have already listed.
Thank you for this interesting comment. Indeed, our SS population is mainly composed of women that could represent a gender bias. Indeed, SS affects predominantly women. In the multicentric French ASSESS cohort, 393 patients with SS were included, with 93.4% of women (Gottenberg JE et al. PLoS One. 2013 May 24;8(5). This percentage is very similar to ours. Then, it seems to be representative of the repartition between men and women with pSS in France. However, pSS in men seems to be more severe with a closer follow-up. It is known that patients who have a closer monitoring have also a better vaccination coverage (Dougados M et al. Ann Rheum Dis. 2015 Sep;74(9):1725-33. doi: 10.1136/annrheumdis-2013-204733. Epub 2014 May 28.). We added this comment in our limitations line 241-246.
Minor concerns:
All tables are of low resolution. Also, make the formatting for all tables uniform.
Thank you for this comment. We provided tables with higher resolution and we made the formatting for all tables uniform.